# Role of Ezrin/Radixin/Moesin in the Surface Localization of Programmed Cell Death Ligand-1 in Human Colon Adenocarcinoma LS180 Cells

**DOI:** 10.3390/ph14090864

**Published:** 2021-08-28

**Authors:** Takuro Kobori, Chihiro Tanaka, Mayuka Tameishi, Yoko Urashima, Takuya Ito, Tokio Obata

**Affiliations:** 1Laboratory of Clinical Pharmaceutics, Faculty of Pharmacy, Osaka Ohtani University, Tondabayashi, Osaka 584-8540, Japan; koboritaku@osaka-ohtani.ac.jp (T.K.); u4117078@osaka-ohtani.ac.jp (C.T.); u4117083@osaka-ohtani.ac.jp (M.T.); urasiyo@osaka-ohtani.ac.jp (Y.U.); 2Laboratory of Natural Medicines, Faculty of Pharmacy, Osaka Ohtani University, Tondabayashi, Osaka 584-8540, Japan; itoutaku@osaka-ohtani.ac.jp

**Keywords:** programmed cell death ligand-1, ezrin, radixin, moesin, colorectal cancer, immune checkpoint inhibitor, post-translational modification

## Abstract

Programmed cell death ligand-1 (PD-L1), an immune checkpoint protein highly expressed on the cell surface in various cancer cell types, binds to programmed cell death-1 (PD-1), leading to T-cell dysfunction and tumor survival. Despite clinical successes of PD-1/PD-L1 blockade therapies, patients with colorectal cancer (CRC) receive little benefit because most cases respond poorly. Because high PD-L1 expression is associated with immune evasion and poor prognosis in CRC patients, identifying potential modulators for the plasma membrane localization of PD-L1 may represent a novel therapeutic strategy for enhancing the efficacy of PD-1/PD-L1 blockade therapies. Here, we investigated whether PD-L1 expression in human colorectal adenocarcinoma cells (LS180) is affected by ezrin/radixin/moesin (ERM), functioning as scaffold proteins that crosslink plasma membrane proteins with the actin cytoskeleton. We observed colocalization of PD-L1 with all three ERM proteins in the plasma membrane and detected interactions involving PD-L1, the three ERM proteins, and the actin cytoskeleton. Furthermore, gene silencing of ezrin and radixin, but not of moesin, substantially decreased the expression of PD-L1 on the cell surface without affecting its mRNA level. Thus, in LS180 cells, ezrin and radixin may function as scaffold proteins mediating the plasma membrane localization of PD-L1, possibly by post-translational modification.

## 1. Introduction

Among global cancer cases, colorectal cancer (CRC) is ranked third in incidence and second in mortality, with an estimated more than 1.9 million new cases and 935,000 deaths worldwide in 2020 [1]. In addition to the high morbidity and mortality in CRC, it appears to be difficult to further advance traditional treatment options, such as surgery, chemotherapy, and radiotherapy [2]. Hence, there is a need to develop new treatment methods to improve the poor prognosis in patients with CRC [3,4,5,6].

The programmed cell death ligand-1 (PD-L1) is an immune checkpoint protein that regulates the immune system by binding to programmed cell death-1 (PD-1). PD-L1 is expressed on the surface of various cell types, including macrophages, dendritic cells, and endothelial cells [7], and is abundantly expressed in a variety of carcinoma cells [7,8,9,10]. PD-L1 helps cancer cells to evade the immune system by preventing T-cell activation and proliferation, mediating T-cell exhaustion, and inducing apoptosis in activated T cells [11]. The disruption of the PD-1/PD-L1 interaction by immune checkpoint therapeutic antibodies (Abs) against PD-1 and PD-L1 reactivates antitumor T-cells, resulting in an antitumor effect [12]. Despite the great success of immune checkpoint therapies blocking the PD-1/PD-L1 interaction in a small subset of various cancer patients [12,13,14,15,16,17,18,19,20], the treatment failed in most cases, including in CRC patients, due to intrinsic unresponsiveness and/or acquired resistance [21,22,23,24,25,26]. Specifically, the early clinical trials demonstrated that patients with CRC appeared to be poor responders to immune checkpoint therapies [27,28,29], indicating an urgent need for novel immunotherapeutic strategies against CRC. Although the high expression of PD-L1 in the cancer tissue region is associated with immune evasion and poor prognosis in CRC patients [29,30,31,32], the mechanisms of unresponsiveness and resistance to PD-1/PD-L1 blockade therapies are largely unknown. Therefore, the identification of potential target molecules that modulate PD-L1 expression in the plasma membrane may satisfy an urgent medical need to develop a novel therapeutic strategy based on boosting the efficacy of PD-1/PD-L1 blockade therapies.

The complex regulation of PD-L1 expression involves a variety of cellular processes, including gene transcription, post-transcriptional and post-translational modifications, and exosomal transport [21,33,34,35]. Because PD-L1 is a transmembrane protein, the protein expression level of PD-L1 in the plasma membrane is extensively regulated by post-translational modifications, such as phosphorylation, glycosylation, ubiquitination, and palmitoylation, all of which affect the localization and function of PD-L1 [21,33,34,36,37]. Therefore, understanding the regulatory mechanisms of PD-L1 expression in the plasma membrane and exploring potential regulators for PD-L1 can lead to the development of new agents that modulate PD-L1 expression in the plasma membrane, which, in turn, may enhance the efficacy of the current immune checkpoint inhibitor therapies.

The members of the ezrin/radixin/moesin (ERM) protein family act as crosslinkers between the actin cytoskeleton and several plasma membrane proteins, such as drug transporters for anticancer agents, including P-glycoprotein (P-gp), multidrug resistance protein (MRP)-2, and MRP-3 [38,39,40,41], as well as other transmembrane proteins involved in cancer progression, including epidermal growth factor receptor 2, several receptor kinases, and cluster of differentiation (CD) 20 [42,43,44]. Interestingly, gene silencing of moesin greatly reduces the plasma membrane localization of PD-L1 in a human breast cancer cell line, implying a novel role of moesin in modulating the protein expression levels of PD-L1 via post-translational modification [45]. However, it remains unclear whether the ERM proteins also regulate the plasma membrane localization of PD-L1 in other cancer cell types.

In this study, we examined the subcellular localization of PD-L1 and the ERM proteins. We also performed RNA interference-mediated gene silencing experiments in LS180 cells, a representative human colorectal adenocarcinoma cell line, to investigate how the ERM proteins affect the gene and protein expression of PD-L1 in the cell surface plasma membrane.

## 2. Results

### 2.1. Gene and Protein Expression Analysis of PD-L1 and the EMR Proteins in LS180 Cells

We initially assessed the gene expression levels of PD-L1, ezrin, radixin, and moesin in LS180 cells and Caco-2 cells which is also a representative human colorectal adenocarcinoma cell line and was used for comparison with LS180 cells. The real-time quantitative reverse transcription-polymerase chain reaction (RT-qPCR) analysis indicated that the gene expression of PD-L1 in LS180 cells was relatively higher than in Caco-2 cells (Figure 1a). In addition, the relative gene expressions of ezrin were approximately equivalent between two cell lines (Figure 1b). By contrast, the gene expression of radixin in LS180 cells was relatively lower than that in Caco-2 cells (Figure 1c). The gene expression of moesin was detected in LS180 cells but not in Caco-2 cells (Figure 1d), the result of which is consistent with previous reports indicating a lack of moesin in Caco-2 cells [46,47]. Thus, we adopted LS180 cells in the subsequent experiments to investigate the involvement of each ERM proteins in the regulatory mechanism of gene and protein expression of PD-L1.

Western blot analysis also showed detectable protein expression levels of PD-L1, in addition to ezrin, radixin, and moesin in whole-cell lysates of LS180 cells (Figure 1e).

### 2.2. Subcellular Localization of PD-L1 and the ERM Proteins in LS180 Cells

Subcellular localization of PD-L1 and the ERM proteins in LS180 cells was assessed by immunofluorescence confocal laser scanning microscopy (CLSM). The fluorescence signal of PD-L1 was highly colocalized with actin, a representative plasma membrane marker (Figure 2a), but not with a nuclear marker (Figure 2b), indicating the preferential localization of PD-L1 in the plasma membrane. The fluorescence signals of all ERM proteins were detected in the plasma membrane where actin was specifically and strongly expressed (Figure 2c,e,g); in contrast, the ERM signals were not merged with those of 4′,6-diamidine-2′-phenylindole dihydrochloride (DAPI), a nuclear marker (Figure 2d,f,h), implying highly specific plasma membrane localization of the ERM proteins. Interestingly, double immunofluorescence staining analysis demonstrated that PD-L1 was highly and specifically colocalized with all ERM proteins in the plasma membrane of LS180 cells (Figure 3a–c).

### 2.3. Molecular Interaction between PD-L1 and the ERM Proteins in LS180 Cells

We next performed the co-immunoprecipitation assay to determine whether PD-L1 interacts with the ERM proteins and the actin cytoskeleton in LS180 cells. Immunoprecipitates from the whole cellular fractions of LS180 cells pulled down with an anti-PD-L1 Ab contained PD-L1 and β-actin combined with high amounts of ezrin and radixin but low amounts of moesin, whereas those pulled down with a control IgG did not contain detectable amounts of any of these proteins (Figure 4). These results implied that PD-L1 intrinsically interacted with all three ERM proteins and the actin cytoskeleton in LS180 cells.

### 2.4. Effect of siRNAs for Each ERM on the Expression Levels of Target mRNAs in LS180 Cells

Small interfering RNAs (siRNA) against each ERM were employed as gene silencing tools to assess the effect of each ERM on the gene and protein expression of PD-L1. The real-time RT-qPCR analysis results indicated that siRNAs against ezrin, radixin, and moesin significantly reduced the mRNA expression level of each target gene compared with that obtained with the transfection reagent alone (Figure 5a–c). Furthermore, the Western blotting analysis showed that the expression of each target protein was also substantially decreased after incubation with siRNAs against ezrin, radixin, and moesin compared with that after incubation with the transfection medium (untreated), the transfection reagent alone, and the nontargeting control siRNA applied at the respective concentration (Figure 5d–f).

### 2.5. Effect of ERM Silencing on Gene and Protein Expression of PD-L1 in LS180 Cells

We then examined how the siRNAs targeting each ERM protein affected the mRNA expression level of PD-L1 to determine the involvement of the ERM proteins in the expression of PD-L1 at the transcriptional level. The PD-L1 mRNA expression level was slightly but not significantly increased by the treatment with siRNA against ezrin and radixin. In contrast, gene silencing of moesin by siRNA significantly increased the PD-L1 mRNA expression (Figure 6a). Notably, the siRNA targeting PD-L1 decreased the mRNA expression of PD-L1 in a dose-dependent manner (Figure 6b). Next, we investigated whether gene silencing of ezrin, radixin, and moesin affected the expression of PD-L1 on the plasma membrane surface of LS180 cells. The flow cytometry analysis results revealed that gene silencing of ezrin and radixin, but not of moesin, significantly decreased the protein expression of PD-L1 on the cell surface to the same level as the gene silencing of PD-L1 (Figure 6c,d). These results implied that ezrin and radixin contributed to the plasma membrane localization of PD-L1, possibly via post-translational modification.

### 2.6. Gene Correlation Analysis of PD-L1 with Ezrin and Radixin in Human Colon Adenocarcinoma

To further study the relationship between PD-L1 and ezrin/radixin in human colorectal tissue, we finally examined whether the gene expression of PD-L1 was correlated with that of ezrin and radixin in colon adenocarcinoma samples from The Cancer Genome Atlas (TGCA) database [48] using UALCAN, a comprehensive and an interactive web resource for in-depth analysis of cancer OMICS data [49]. Gene correlation analysis of colon adenocarcinoma samples from TGCA detected a positive correlation between PD-L1 and ezrin (Pearson correlation coefficient: 0.36) or moesin (Pearson correlation coefficient: 0.42) (Figure 7a,b), but no correlation between PD-L1 and radixin.

## 3. Discussion

In this study, we initially determined the gene and protein expression levels of ezrin, radixin, and moesin in LS180 cells, which were in line with our recent study [47]. Moreover, the present CLSM analysis data clearly demonstrated that all three ERM proteins were preferentially and highly localized to the plasma membrane. Other researchers have reported that human colon adenocarcinoma sublines (EB3, 3LNLN, and 5W) shows higher protein expression levels of ezrin in the whole cellular extracts with higher migratory and adhesive abilities in comparison with their parental LS180 cells [50]. Moreover, radixin is found to be significantly increased in human colon tumor tissue [51] and almost equally upregulated in various kinds of human CRC cell lines, such as NCM460, HT-29, Caco-2, HCT116, and LoVo cells [52]. In contrast, there are few reports showing the gene and protein expressions of moesin in human colon adenocarcinoma cells [47]. Although the expression profile of ERM proteins is dependent on the human CRC cell types, the present and previous findings showed that LS180 cells, which carries the genes and proteins for all three ERM, has the potential utility to assess the role of each ERM proteins in the regulatory mechanism of PD-L1 expression in human colon adenocarcinoma cells.

We also detected the gene and protein expression of PD-L1 in LS180 cells, which was similar with previous observations indicating the protein expression of PD-L1 in human CRC cell lines HCT116, LOVO, and HCT15 [53,54]. Furthermore, our CLSM analysis demonstrated, for the first time, the specific distribution of all three ERM proteins and PD-L1 in the plasma membrane of LS180 cells due to colocalization of PD-L1 with all three ERM proteins. Taken together, gene and protein expression of PD-L1 and the ERM proteins in LS180 cells results in a highly colocalized distribution of these proteins in the plasma membrane.

Cumulative evidence suggests that the protein expression level and functional activity of plasma membrane proteins do not necessarily depend on the transcriptional process [38,55,56,57]. Recently, two research groups have independently identified the chemokine-like factor-like MARVEL transmembrane domain-containing protein 6 (CMTM6) as a crucial regulator of PD-L1 in a broad range of cancer cells; they found that CMTM6 promotes PD-L1 stabilization in the plasma membrane by inhibiting the ubiquitination of PD-L1, which prevents its subsequent degradation via lysosomes [36,37]. Interestingly, CMTM6 enhances the PD-L1 protein pool without affecting the PD-L1 transcription levels and interacts with PD-L1 at the cell surface, resulting in reduced ubiquitination and prolonged half-life of the PD-L1 protein [36]. The ERM proteins are scaffold proteins involved in post-translational modification, and they regulate the subcellular localization of several cancer-related proteins by anchoring them to the plasma membrane in cancer cells [38,39,40,42,43]. Furthermore, the fact that the phosphorylated ERM proteins also colocalize with the T cell receptor (TCR) αβ, a member of the immunoglobulin superfamily, and the actin filaments [58] appears to indicate that crosslinking the TCR complex to the actin cytoskeleton is a novel function of the ERM proteins. Meng et al. recently used co-immunoprecipitation assays to demonstrate that moesin physiologically interacts with PD-L1 and is necessary for the stabilization of PD-L1 on the cell surface of human breast cancer cells [45]. Our immunoprecipitation assay demonstrated, for the first time, that the physiological interaction between PD-L1 and the ERM proteins involved higher amounts of ezrin and radixin but lower amounts of moesin. We further observed that the immunoprecipitates contained tripartite complexes consisting of PD-L1, ERM, and the actin cytoskeleton. The shift in the molecular weight of all three proteins in the immunoprecipitates may reflect, at least in part, the phosphorylated status of ERM proteins bound to PD-L1 in the plasma membrane, although the details remain to be determined. Collectively, the present and previous findings allow us to hypothesize that all three ERM proteins may serve as scaffold proteins that regulate the plasma membrane localization of PD-L1 via post-translational modifications in LS180 cells.

To investigate the effect of each ERM on the gene and protein expression of PD-L1 in LS180 cells, we employed siRNAs to silence each ERM gene separately. We confirmed that each of the siRNAs targeting ezrin, radixin, or moesin substantially suppressed the respective target mRNA and protein in LS180 cells. The siRNA concentrations used in this study never affected the viability of LS180 cells, as determined in our previous report [47]. Interestingly, the cell surface expression of PD-L1 was significantly decreased by the knockdown of ezrin and radixin, whereas the moesin knockdown had no effect. Nevertheless, none of the siRNAs reduced the PD-L1 mRNA expression. Using an in vitro cell culture model, Meng et al. also discovered that gene silencing of moesin substantially suppressed PD-L1 expression on the cell surface without any impact on the PD-L1 mRNA level, triggering T cell activation in this model [45]; however, the effect of ezrin and radixin gene suppression remains to be examined. There is cumulative evidence that the ERM protein expression profile differs among cancer cell types [41] and that each ERM protein differently contributes to the plasma membrane localization of P-gp, a typical partner protein for ERM, depending on the cancer cell type, organ, and animal species [38,47,59,60,61,62].

Moreover, the gene correlation analysis revealed a positive correlation of PD-L1 with ezrin and moesin but not radixin in the clinical human colon adenocarcinoma samples from the TGCA database. Recent clinical retrospective study found that the positive PD-L1 expression rate in the CRC tissues was significantly higher in patients with lymph node metastasis than in those without lymph node metastasis, and also increased gradually as the cancer stage became more advanced [32]. Furthermore, emerging evidence has demonstrated that during cancer development, the protein expressions of ezrin especially in the plasma membrane of cancer cells are elevated, leading to cancer progression, invasion, and metastasis [63,64,65]. These observations raise the possibility that differential expression of ERM proteins and PD-L1 may exists during cancer development. Collectively, the discrepancies between the basic and clinical studies may be, at least in part, due to the different ERM and PD-L1 expression profiles that depend on the cancer cell types and the clinical cancer stages.

Surprisingly, gene silencing of moesin significantly increased the PD-L1 mRNA level. The pro-inflammatory cytokines, such as interferon (IFN)-ɤ, tumor necrosis factor (TNF)-α, and interleukin (IL)-6, are well known to be key inducers for PD-L1 expression at the transcriptional level in tumor tissues [21]. In fact, moesin siRNA significantly increased the mRNA expression levels of TNF and IL-6 in LS180 cells, although those of IFN-ɤ were undetectable (Appendix A). Some reports have already indicated that the neutralization of moesin by an anti-moesin Ab can induce the secretion of IFN-ɤ, TNF-α, and IL-6 in T-cells, neutrophils, and adhesive monocytes isolated from human peripheral whole blood [66,67]. Additionally, high expression levels of serum IFN-ɤ and TNF-α were observed in patients with myeloperoxidase-antineutrophil cytoplasmic antibody-associated vasculitis who were positive for anti-moesin auto-Ab [67]. Although the cell types were different, these previous observations could partly explain our present results that the knockdown of moesin increased the PD-L1 mRNA level in LS180 cells. The detailed mechanism remains to be elucidated and should be addressed in future studies.

In summary, although we observed colocalization and interactions of PD-L1 with all three ERM proteins in the plasma membrane, the ERM proteins differently affected the gene and protein expression of PD-L1 via transcriptional and/or post-translational processes. In LS180 cells, ezrin and radixin may, at least in part, function as essential scaffold proteins mediating the plasma membrane localization of PD-L1 (Figure 8).

## 4. Materials and Methods

### 4.1. Cell Culture

The human colon adenocarcinoma cell lines, LS180 and Caco-2, were purchased from the European Collection of Cell Cultures (ECACC) (EC87021202-F0 and EC86010202-F0; KAC, Hyogo, Japan). LS180 cells were cultured in Dulbecco’s modified Eagle’s medium (DMEM) containing 1500 mg/L glucose (FUJIFILM Wako Pure Chemical) and Caco-2 cells were cultured in DMEM with 4500 mg/L glucose (FUJIFILM Wako Pure Chemical). In both the cases, the medium was supplemented with heat-inactivated 10% fetal bovine serum (BioWest, Nuaillé, France). The cultures were maintained at 37 °C in a humidified atmosphere with 5% CO_2_.

### 4.2. Transfection of Cells with siRNAs

LS180 cells were cultured until 70–80% confluent in flasks. Then, they were seeded at a density of 2.5 × 10^4^ cells/well in 24-well cell culture plates (Corning, Glendale, AZ, USA) for total RNA isolation and at a density of 1.0 × 10^5^ cells/well in 6-well cell culture plates (Corning) for total protein isolation. The cultures were incubated overnight at 37 °C in a humidified atmosphere with 5% CO_2_ to allow for attachment. Silencer Select siRNAs for each target gene (Thermo Fisher Scientific, Tokyo, Japan), which selectively suppress the target gene expression, were diluted with Opti-MEM (Thermo Fisher Scientific). Then, cells were transfected with siRNAs targeting human ezrin (5 nM), radixin (5 nM), moesin (2 M), or PD-L1 (5 or 10 nM) used at specific concentrations in the presence of the Lipofectamine RNAiMAX Transfection Reagent (Thermo Fisher Scientific). The transfection reagent volume was 1.0 µL/well for total RNA isolation or 4.0 µL/well for total protein isolation. The concentration of each siRNA and transfection reagent volume were determined to have a high knockdown activity and low cytotoxicity as shown in our recent publication [47] and Appendix A.

After adding the siRNA and transfection reagent, cells were continuously cultured for 3 days (RNA isolation) or 4 days (protein isolation and flow cytometry analysis) without exchanging medium. Each treatment period adopted in this study was determined based on the manufacture’s protocol. Silencer Select nontargeting control siRNA (Thermo Fisher Scientific), which has no significant similarity to human gene sequences and has minimal effects on gene suppression, was used as a nontargeting control for each siRNA.

### 4.3. RNA Isolation and Real-Time RT-qPCR

Total RNA was extracted from LS180 cells and Caco-2 cells using ISOSPIN Cell & Tissue RNA (NIPPON GENE, Tokyo, Japan) according to the manufacturer’s protocol. Total RNA purity and quantity were evaluated using a Nano Drop ND-1000 spectrophotometer (Thermo Fisher Scientific). The extracted total RNA preparations were analyzed on 96-plates with the CFX Connect Real-Time PCR Detection System (Bio-Rad Laboratories, Tokyo, Japan) using RNA concentrations in the range of 10–50 ng/well as the templates for real-time RT-qPCR amplification. The reactions were performed with the One Step TB Green Prime Script PLUS PT-PCR Kit (Takara Bio, Shiga, Japan) and the gene-specific primer for human PD-L1, ezrin, radixin, moesin, IFN-ɤ, TNF, IL-6, and β-actin (all purchased from Takara Bio) at a final concentration of 0.4 µM. The reaction program had a reverse transcription reaction step at 42 °C for 5 min, and a denaturation step at 95 °C for 10 s, subsequently followed by 40–45 cycles of denaturation at 95 °C for 5 s and annealing/extension at 60 °C for 30 s. The relative mRNA levels of the target genes were normalized to those of β-actin amplified as an internal reference from the same sample and computed according to the comparative quantification cycle (Cq) method (2^−ΔΔCq^) using the Bio-Rad CFX Manager software version 3.1 (Bio-Rad Laboratories). The sequences of gene-specific RT-qPCR primers are shown in Table 1.

### 4.4. CLSM Analysis

CLSM analysis was carried out as described previously with some modifications [47,68,69]. Single and double immunofluorescence staining steps were performed to confirm the subcellular localization of PD-L1, ezrin, radixin, and moesin, as along with the colocalization of PD-L1 with ezrin, radixin, and moesin.

#### 4.4.1. Single Immunofluorescence Staining

LS180 cells were seeded at a density of 0.5 × 10^5^ cells on a polylysine-coated 35 mm glass-bottom dish with an inner diameter of 14 mm (Matsunami Glass, Osaka, Japan). Incubation was performed overnight at 37 °C under humidified conditions with 5% CO_2_ to allow for attachment. The cells were washed with Dulbecco’s phosphate saline (D-PBS) (FUJIFILM Wako Pure Chemical) and fixed with 4% paraformaldehyde (PFA) (FUJIFILM Wako Pure Chemical) at room temperature for 15 min, followed by washing thrice with D-PBS. Subsequently, 0.5% Triton-X100 (Thermo Fisher Scientific) was added and incubated at room temperature for 15 min to enhance the permeability of the plasma membrane. Next, to block non-specific protein–protein interactions, the cells were incubated in a blocking buffer containing D-PBS, supplemented with 10% normal goat serum (Thermo Fisher Scientific), 1% bovine serum albumin (BSA) (FUJIFILM Wako Pure Chemical), and 0.1% Tween-20 (Nacalai Tesque, Kyoto, Japan), at room temperature for 60 min. To determine the intracellular localization of PD-L1, cells were incubated overnight at 4 °C in the dark under humidified conditions with an Alexa Fluor 488-conjugated rabbit anti-PD-L1 Ab (25048; Cell Signaling Technology, Danvers, MA, USA) at a dilution of 1:50 in blocking buffer. After washing thrice with D-PBS supplemented with 0.1% Tween-20 (PBS-T), the nuclei or plasma membrane were counterstained with a NucRed Live 647 ReadyProbes Reagent (Thermo Fisher Scientific) or an ActinRed 555 ReadyProbes Reagent (Thermo Fisher Scientific), respectively, in blocking buffer for 30 min at room temperature. The cells were washed thrice with PBS-T, and then the Fluoro-KEEPER Antifade Regent Non-Hardening Type (Nacalai Tesque) was added for storage and prevention of quenching.

To observe the intracellular localization of ezrin, radixin, or moesin, cells were processed in preparation for the Ab incubation, as described above. The cells were incubated overnight at 4 °C in the dark under humidified conditions with a rabbit anti-ezrin Ab (3145s; Cell Signaling Technology) at a dilution of 1:50, a rabbit anti-radixin Ab (GTX105408; Gene Tex, Alton Pkwy Irvine, CA, USA) at a dilution of 1:50, or a rabbit anti-moesin Ab (3150s; Cell Signaling Technology) at a dilution of 1:25 in blocking buffer. After washing thrice with PBS-T, the cells were incubated for 60 min at room temperature with an Alexa Fluor 488-conjugated goat anti-rabbit IgG (H+L) Ab (R37116; Thermo Fisher Scientific) or an Alexa Fluor 594-conjugated goat anti-rabbit IgG (H+L) Ab (R37117; Thermo Fisher Scientific) both prepared as dilutions by adding by adding approximately 40 µL to 460 µL blocking buffer. After washing thrice with PBS-T, the plasma membranes were counterstained with an Actin Red 555 ReadyProbes Reagent (Thermo Fisher Scientific) in the blocking buffer for 30 min at room temperature. The cells were washed thrice with PBS-T, and then the Fluoro-KEEPER Antifade Regent Non-Hardening Type (Nacalai Tesque) was added for storage and prevention of quenching. Similarly, nuclei were counterstained with Fluoro-KEEPER Antifade Reagent containing DAPI (Nacalai Tesque). The preserved cells were observed and photographed at 0.5–0.7 µm intervals on the z-axis at an original magnification of 60× with a Nikon Al confocal laser microscope system (Nikon Instrument, Tokyo, Japan). The two- or three-dimensional images were reconstructed from the acquired pictures using the NIS-Elements Ar Analysis software (Nikon Instruments).

#### 4.4.2. Double Immunofluorescence Staining

The cells were processed in preparation for the Ab incubation, as described above. The cells were incubated overnight at 4 °C in the dark under humidified conditions with a rabbit anti-ezrin Ab at a dilution of 1:50, a rabbit anti-radixin Ab at a dilution of 1:50, or a rabbit anti-moesin Ab at a dilution of 1:25 in blocking buffer. After washing thrice with PBS-T, the cells were incubated for 60 min at room temperature with an Alexa Fluor 594-conjugated goat anti-rabbit IgG (H+L) (R37117; Thermo Fisher Scientific) prepared as dilution by adding by adding approximately 40 µL to 460 µL blocking buffer. Subsequently, the cells were washed thrice with PBS-T and incubated overnight at 4 °C in the dark under humidified conditions with an Alexa Fluor 488-conjugated rabbit anti-PD-L1 Ab (25048; Cell Signaling Technology). After washing thrice with PBS-T, Fluoro-KEEPER Antifade Regent Non-Hardening Type was added for storage and prevention of quenching. The preserved cells were observed and photographed at 0.5–0.7 µm intervals for z-axis at an original magnification of 60× with a Nikon Al confocal laser microscope system.

### 4.5. Flow Cytometry Analysis

Flow cytometry analysis was carried out as described previously with some modifications [47]. After treatment of LS180 cells with siRNAs for 4 days, the cells were detached using 500 μL of Accutase (Nacalai Tesque) and transferred into 5 mL tubes filled with 2 mL of D-PBS for centrifugation (260× *g* for 5 min at 4 °C). Subsequently, the cells were incubated with an Alexa Fluor 488-conjugated rabbit anti-PD-L1 Ab (Cell Signaling Technology, Danvers, MA, USA) in a volume of 3 μL/tube in a labeling buffer consisting of D-PBS supplemented with 5% normal horse serum (Biowest) and 1% sodium azide (FUJIFILM Wako Pure Chemical) for 60 min at 4 °C. After rinsing in the labeling buffer and centrifugation (260× *g* for 5 min at 4 °C), the cell pellet was resuspended in 600 µL of D-PBS containing propidium iodide (PI) (Dojindo Laboratories, Kumamoto, Japan) at a concentration of 2 µg/mL to exclude PI-positive dead cells. Thereafter, the cells were analyzed with a Cell Analyzer EC800 (Sony Imaging Products & Solutions, Tokyo, Japan). Data were processed using the EC800 Analysis software (Sony Imaging Products & Solutions) to determine the mean fluorescence intensity of the Alexa Fluor 488-PD-L1 on the cell surface of LS180 cells.

### 4.6. Protein Isolation

After treatment of cells with siRNAs for 4 days without exchanging medium, cells were rinsed twice with ice-cold D-PBS and subsequently lysed in radio-immunoprecipitation assay (RIPA) buffer containing a protease inhibitor cocktail for 30 min on ice. The cell debris was removed by centrifugation (15,000× *g*, 4 °C for 10 min), and then the supernatant of the resulting suspension was collected as the total cell lysate. The protein concentration was quantified using a TaKaRa BCA Protein Assay Kit (Takara Bio).

### 4.7. Western Blotting

Western blotting was conducted as described previously with some modifications [68,69]. Briefly, to perform sodium dodecyl sulfate (SDS)-polyacrylamide gel electrophoresis (PAGE), total lysates of LS180 cells were diluted with equal volumes of Sample Buffer Solution (2×), which contained 0.125 M Tris–HCl, 4% SDS, 20% glycerin, 0.01% bromophenol blue, and 10% 2-mercaptoethanol (Nacalai Tesque), and boiled at 97 °C for 5 min. Equal protein amounts ranging from 2.5 to 16.0 µg/lane, depending on target proteins, were loaded and separated via SDS-PAGE, which was followed by the electrotransfer onto a nitrocellulose membrane (Bio-Rad Laboratories). Then, the blotting consistency was determined with Ponceau S (MP Biomedicals, Santa Ana, CA, USA) staining. The membrane was incubated in the blocking buffer containing 5% non-fat dry milk (FUJIFILM Wako Pure Chemical) in PBS-T for 60 min at room temperature. Subsequently, the membrane was probed with rabbit Abs against ezrin (3145s; Cell Signaling Technology) at a dilution of 1:1000, radixin (GTX105408; Gene Tex) at a dilution of 1:1000, moesin (3150s; Cell Signaling Technology) at a dilution of 1:1000, a horse radish peroxidase (HRP)-conjugated rabbit Ab against PD-L1 (51296s; Cell Signaling Technology) at a dilution of 1:1000, or a mouse Ab against glyceraldehyde-3-phosphate dehydrogenase (GAPDH) (MAB374; Merck) at a dilution of 1:20,000 used as an internal control, in blocking buffer at 4 °C overnight. Blots were then washed with PBS-T and incubated with HRP-conjugated secondary Abs against an anti-rabbit IgG (5220-0336; SeraCare Life Sciences, Milford, MA, USA) at a dilution of 1:5000 for ezrin, radixin, and moesin, or an anti-mouse IgG (5220-0341; SeraCare Life Sciences) at a dilution of 1:10,000 for GAPDH in blocking buffer for 60 min at room temperature. After washing with PBS-T, the immune complexes were visualized using the Pierce ECL Western Blotting Substrate (Thermo Fisher Scientific). The chemiluminescence signal intensities of the immune reactive bands were detected and analyzed using the Light Capture instrument (ATTO, Tokyo, Japan) with the Image Analysis Software CS Analyzer (ATTO). All the original Western blotting images are shown in Appendix A.

### 4.8. Immunoprecipitation Assay

The immunoprecipitation assay was conducted as described previously [70,71] with some modifications. Briefly, 500 μL of the total whole-cell lysate processed as described above was incubated with 50 μL of nProtein A Sepharose 4 Fast flow (Cytiva, Tokyo, Japan) for 60 min at 4 °C on a rotating wheel to remove non-specific binding proteins to nProtein A Sepharose. After nProtein A Sepharose was pelleted via centrifugation (3000× *g*, 4 °C for 1 min), the pre-cleaned supernatants of the whole-cell lysates were incubated overnight at 4 °C on a rotating wheel with a rabbit Ab against PD-L1 (13684s; Cell Signaling Technology) or its isotype control Ab (3900s; Cell Signaling Technology), both at a dilution of 1:30. Then, 50 μL of nProtein A Sepharose was added to the lysate and subsequently incubated at 4 °C for 3 h on a rotating wheel. The precipitates were rinsed thrice with RIPA buffer containing protease inhibitor cocktails, followed by centrifugation (3000× *g*, 4 °C for 1 min) to obtain the immunoprecipitated pellets. After resuspension of the immunoprecipitated pellets in the sample buffer solution (2×) for SDS-PAGE (Nacalai Tesque), the pellets were boiled at 97 °C for 5 min and pelleted by centrifugation (15,000× *g*, 4 °C for 1 min). The supernatant fractions and total cell lysates (input) were adjusted to protein amounts ranging from 0.20 to 3.0 µg/lane, depending on the target protein, and separated via SDS-PAGE, which was followed by the electrotransfer onto a nitrocellulose membrane (Bio-Rad Laboratories). Subsequently, Western blotting and analysis of chemiluminescence signals were performed as described in Section 4.7. Then, consistent blotting was checked by Ponceau S (MP Biomedicals) staining. The membrane was incubated in the blocking buffer containing 5% non-fat dry milk (FUJIFILM Wako Pure Chemical) for PD-L1, moesin, and β-actin or 5% BSA (FUJIFILM Wako Pure Chemical) for ezrin and radixin in PBS-T for 60 min at room temperature. Subsequently, the membrane was probed with rabbit Abs against ezrin (3145s; Cell Signaling Technology) at a dilution of 1:1000, radixin (GTX105408; Gene Tex) at a dilution of 1:1000, moesin (3150s; Cell Signaling Technology) at a dilution of 1:1000, a mouse Ab against β-actin (A1978; Merck) at a dilution of 1:10,000 or HRP-conjugated rabbit Ab against PD-L1 (51296s; Cell Signaling Technology) at a dilution of 1:1000 in respective blocking buffer at 4 °C overnight. Blots were then washed with PBS-T and incubated with HRP-conjugated secondary Abs against a rabbit IgG (5220-0336; SeraCare Life Sciences) at a dilution of 1:5000 for ezrin, radixin, and moesin or a mouse IgG (5220-0341; SeraCare Life Sciences) at a dilution of 1:10,000 for β-actin in respective blocking buffer for 60 min at room temperature. After washing with PBS-T, the immune complexes were visualized using the Pierce ECL Western Blotting Substrate (Thermo Fisher Scientific). The chemiluminescence signal intensities of the immune reactive bands were detected and analyzed using the Light Capture instrument (ATTO) with the Image Analysis Software CS Analyzer (ATTO).

### 4.9. Statistical Analysis

Data are presented as means ± standard error of the mean (SEM). Statistical analysis was performed using the Prism version 3 software (GraphPad Software, La Jolla, CA, USA). Statistical significance was assessed by performing the one-way analysis of variance (ANOVA) followed by Dunnett’s test for multiple comparisons. Differences with *p* values < 0.05 were considered significant.

## 5. Conclusions

In the present study, we detected PD-L1 mRNA and protein expression in LS180 cells. The PD-L1 protein was specifically localized in the plasma membrane. Furthermore, colocalization and interactions of PD-L1 occurred with all three ERM proteins. Gene silencing experiments revealed that ezrin and radixin functioned as scaffold proteins that mediated the plasma membrane localization of PD-L1, possibly by post-translational modification without affecting PD-L1 transcription. Thus, PD-L1 expression modulation in the plasma membrane of cancer cells by therapeutic agents targeting ezrin and/or radixin may represent a novel treatment strategy to boost the efficacy of PD-1/PD-L1 blockade therapies in CRC. This novel treatment strategy might give rise to possible adjuvant therapies for the current PD-1/PD-L1 blockade Abs, offering significant therapeutic benefits to CRC patients with intrinsic unresponsiveness and/or acquired resistance to the immune checkpoint therapies at present.

## Figures and Tables

**Figure 1 pharmaceuticals-14-00864-f001:**
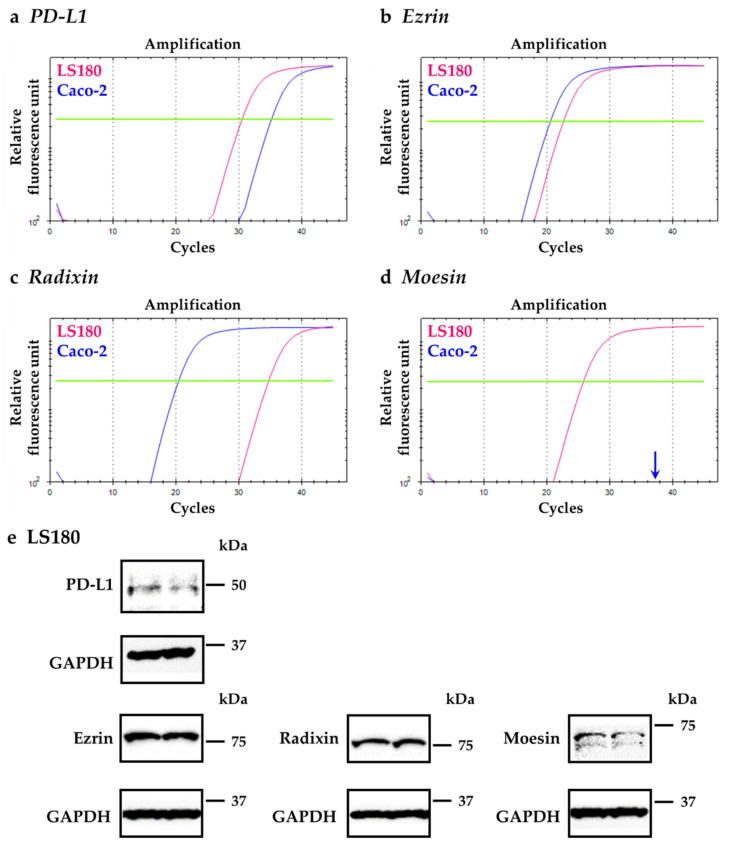
Gene and protein expression profiles of programmed cell death ligand-1 (PD-L1), ezrin, radixin, and moesin (ERM) in LS180 cells and Caco-2 cells. (**a**–**d**) Representative amplification curves of (**a**) PD-L1, (**b**) ezrin, (**c**) radixin, and (**d**) moesin mRNA expressions in LS180 cells (pink line) and Caco-2 cells (blue line) as determined by real-time quantitative reverse transcription-polymerase chain reaction (RT-qPCR). (**e**) Representative images of Western blots for detection of PD-L1, ezrin, radixin, moesin, and glyceraldehyde-3-phosphate dehydrogenase (GAPDH) proteins in whole-cell lysates of LS180 cells. Molecular weights are indicated in kDa. The data are representative of three independent experiments using at least three independent samples of total RNA and protein extracts.

**Figure 2 pharmaceuticals-14-00864-f002:**
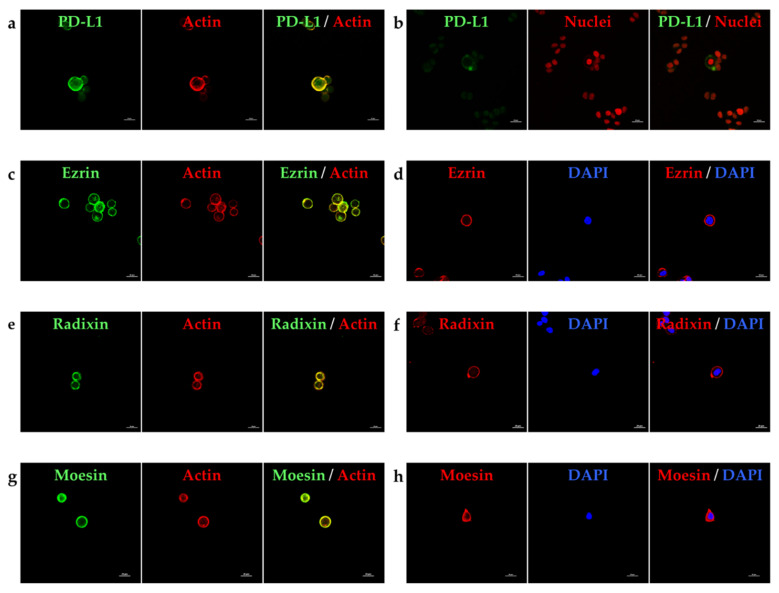
Subcellular localization of programmed cell death ligand-1 (PD-L1), ezrin, radixin, and moesin detected in LS180 cells by confocal laser scanning microscopy. In a three-dimensional reconstruction of optically sectioned LS180 cells, PD-L1 (green) was preferentially colocalized with (**a**) actin (red) on the plasma membrane but not with (**b**) nuclei (red). In the same sections of LS180 cells, (**c**) ezrin, (**e**) radixin, and (**g**) moesin (green) were distributed near the plasma membrane and preferentially colocalized with actin (red) used as a plasma membrane marker. Additionally, (**d**) ezrin, (**f**) radixin, and (**h**) moesin (red) were not colocalized with 4′,6-diamidine-2′-phenylindole dihydrochloride (DAPI), a typical nuclear marker (blue). Scale bars: 20 μm. All images are representative of at least three independent experiments.

**Figure 3 pharmaceuticals-14-00864-f003:**
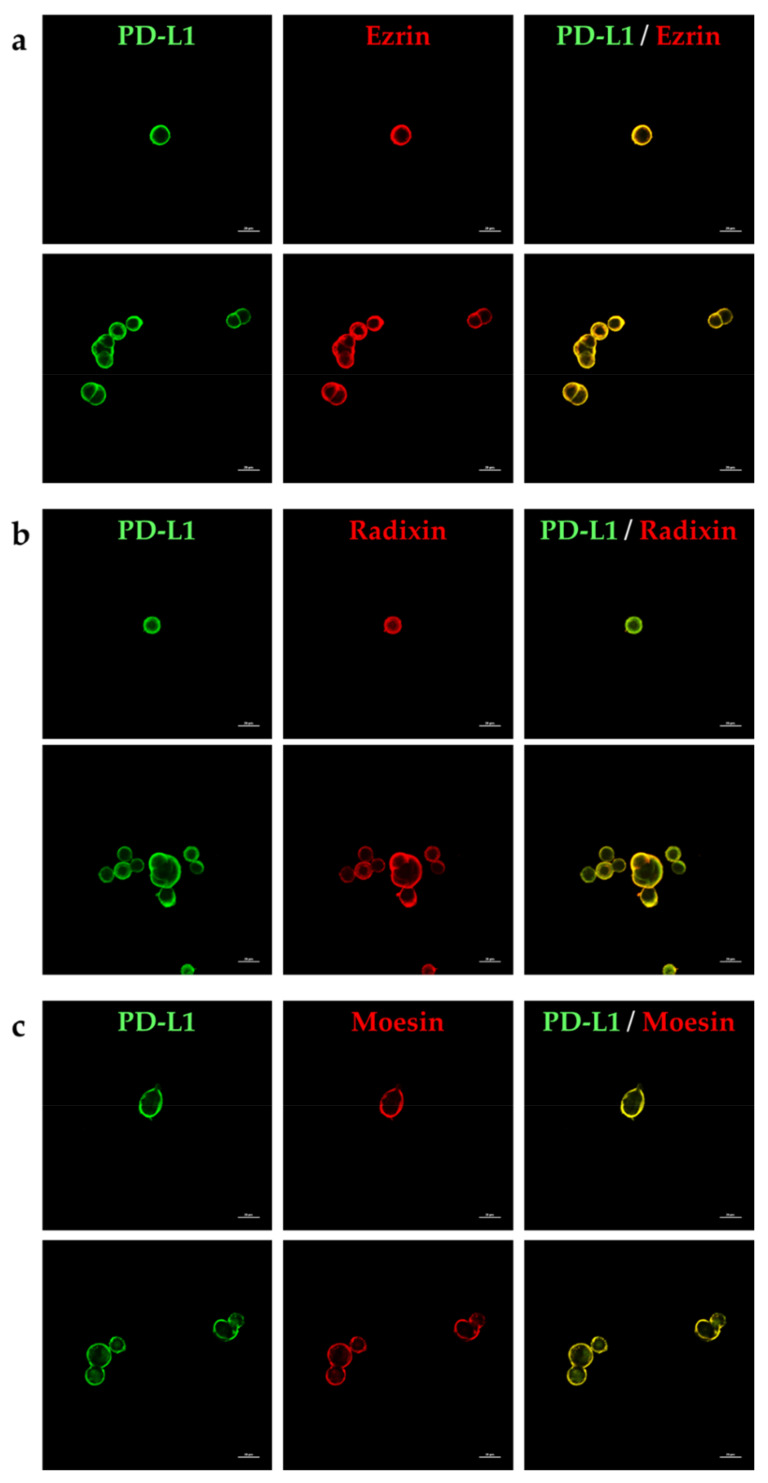
Colocalization of programmed cell death ligand-1 (PD-L1) with ezrin, radixin, and moesin in the plasma membrane of LS180 cells. In a three-dimensional reconstruction of optically sectioned LS180 cells obtained by confocal laser scanning microscopy analysis, PD-L1 (green) was highly colocalized with (**a**) ezrin, (**b**) radixin, and (**c**) moesin (red) on the plasma membrane. Scale bars: 20 μm. All images are representative of at least three independent experiments.

**Figure 4 pharmaceuticals-14-00864-f004:**
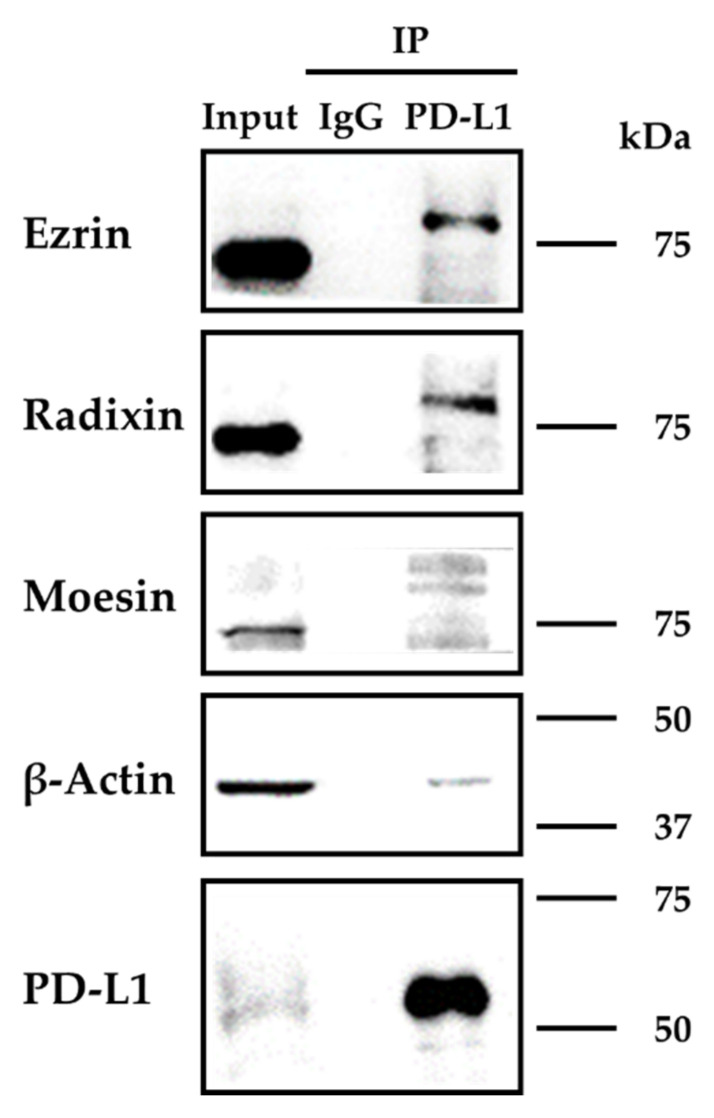
Immunoprecipitation analysis to detect the protein–protein interactions between programmed cell death ligand-1 (PD-L1) and ezrin, radixin, and moesin (ERM) in LS180 cells. The whole-cell lysates of LS180 cells were immunoprecipitated with an anti-PD-L1 antibody or its isotype control antibody. Images of representative Western blots probed for the presence of ezrin, radixin, moesin, β-actin, and PD-L1 in the whole-cell lysates (input) and the immunoprecipitates (IP) pulled down with a control IgG or an anti-PD-L1 antibody. Molecular weights are indicated in kDa. The images are representative of three independent immunoprecipitation experiments using at least three independent replicates of protein extracts.

**Figure 5 pharmaceuticals-14-00864-f005:**
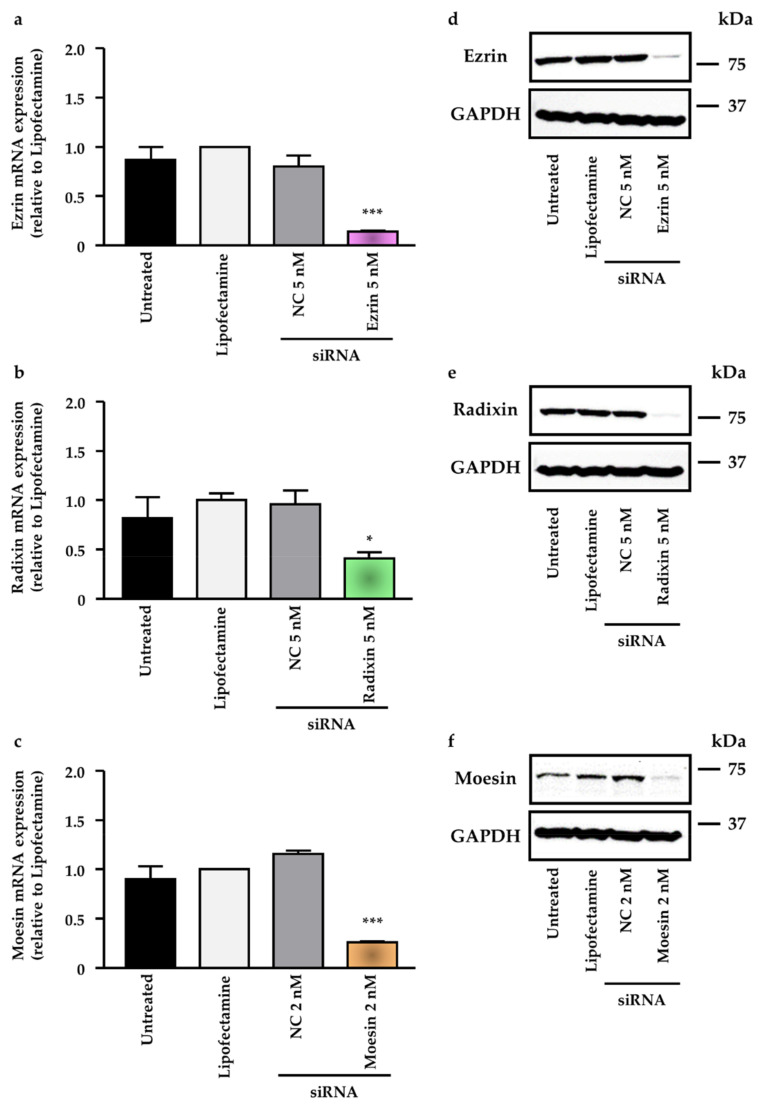
Effect of siRNAs targeting ezrin, radixin, and moesin (ERM) on the mRNA and protein expression of each target gene in LS180 cells. Cells were incubated with the transfection medium (Untreated), transfection reagent (Lipofectamine), nontargeting control (NC) siRNA, and specific siRNAs for ezrin, radixin, or moesin and then cultured for 3 days to measure the mRNA expression or 4 days to measure the protein expression. (**a**–**c**) Expression level of each mRNA normalized with β-actin in cells treated with siRNAs relative to that in cells treated with the transfection reagent alone was measured by real-time quantitative reverse transcription-polymerase chain reaction. *n* = 3, *** *p* < 0.001, * *p* < 0.05 vs. Lipofectamine. All data were expressed as the mean ± SEM and analyzed by one-way ANOVA followed by Dunnett’s test. (**d**–**f**) Images of representative Western blots probed for the presence of ezrin, radixin, moesin, and the control protein glyceraldehyde-3-phosphate dehydrogenase (GAPDH) in whole-cell lysates of LS180 cells. Molecular weights are indicated in kDa. The data are representative of three independent experiments using at least three independent replicates of protein extracts.

**Figure 6 pharmaceuticals-14-00864-f006:**
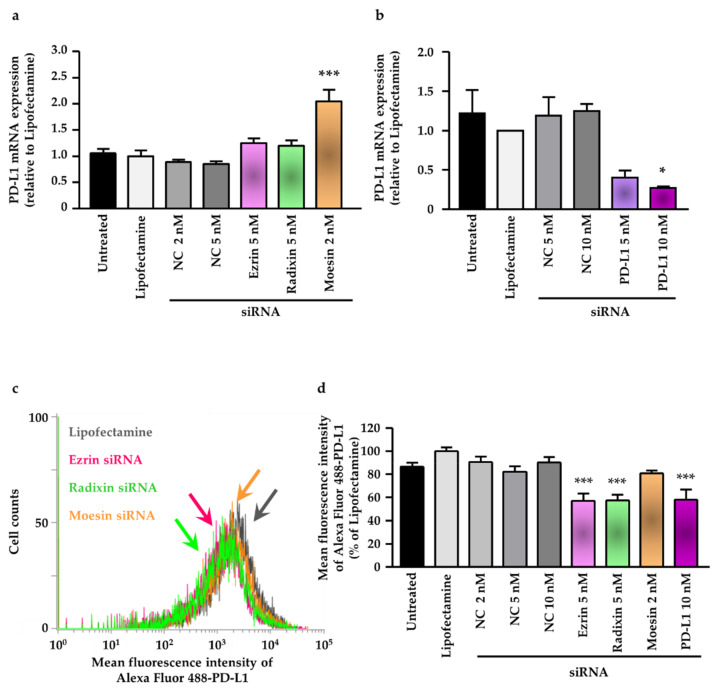
Effect of siRNAs targeting ezrin, radixin, moesin, or the programmed cell death ligand-1 (PD-L1) on the mRNA and plasma membrane expression of PD-L1 in LS180 cells. Cells were incubated with the transfection medium (Untreated), transfection reagent (Lipofectamine), nontargeting control (NC) siRNA, and specific siRNAs for ezrin, radixin, moesin, or PD-L1 and then cultured for 3 days to measure the mRNA expression or 4 days to determine the protein expression on the cell surface. (**a**,**b**) The mRNA expression level of PD-L1 normalized with β-actin in cells treated with each siRNA relative to that in cells treated with Lipofectamine alone was determined by real-time quantitative reverse transcription-polymerase chain reaction; (**a**) *n* = 3, *** *p* < 0.001 vs. Lipofectamine; (**b**) *n* = 3, * *p* < 0.05 vs. Lipofectamine. All data were expressed as the mean ± SEM and analyzed by one-way ANOVA followed by Dunnett’s test. (**c**) An overlay of the representative histograms for the mean fluorescence intensity of Alexa Fluor 488-labeled PD-L1 on the plasma membrane of LS180 cells treated with Lipofectamine (gray line), ezrin siRNA (pink line), radixin siRNA (green line), and moesin siRNA (orange line) as measured by flow cytometry. (**d**) The calculated mean fluorescence intensities of PD-L1 relative to Lipofectamine alone on the plasma membrane surface are shown for all the treatments; *n* = 4, *** *p* < 0.001 vs. Lipofectamine. All data were expressed as the mean ± SEM and analyzed by one-way ANOVA followed by Dunnett’s test.

**Figure 7 pharmaceuticals-14-00864-f007:**
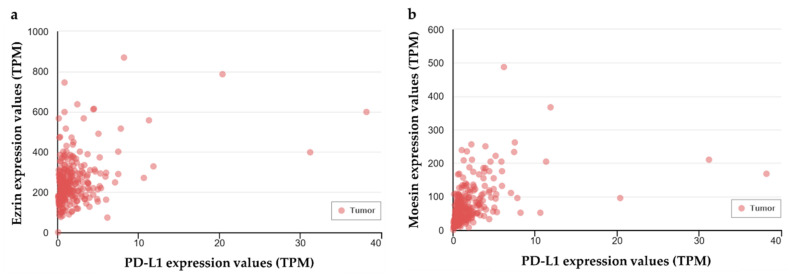
Positive correlation of gene expression between programmed cell death ligand-1 (PD-L1) with ezrin and moesin in human colon adenocarcinoma samples. The correlation analysis of the gene expression of PD-L1 and (**a**) ezrin or (**b**) moesin was performed in clinical colon adenocarcinoma samples (*n* = 286) from The Cancer Genome Atlas (TGCA) database using the publicly available online tool UALCAN. The estimated gene expression values based on RNA-seq data are presented as transcripts per million (TPM).

**Figure 8 pharmaceuticals-14-00864-f008:**
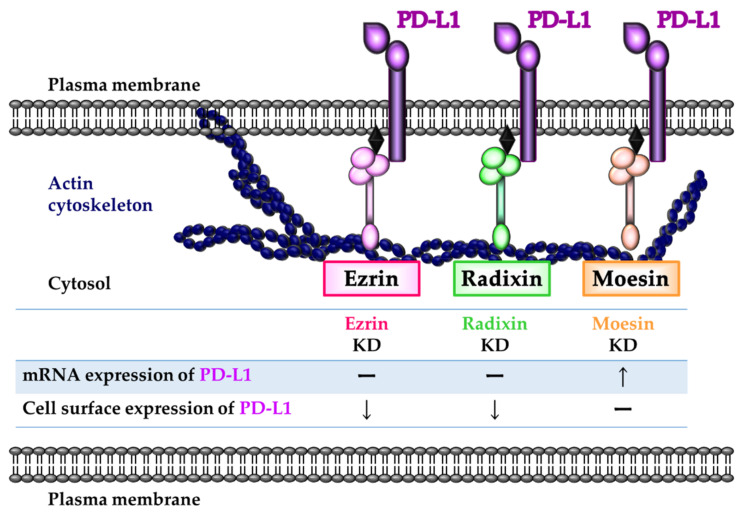
Schematic representation illustrating the involvement of each ERM in the regulation of PD-L1 expression in LS180 cells. Ezrin and radixin contribute to the plasma membrane localization of PD-L1 by crosslinking with the actin cytoskeleton, possibly serving as scaffold proteins without any effect on the transcriptional level. In contrast, moesin negatively regulates PD-L1 expression at the mRNA level and interacts with the PD-L1 protein. Therefore, among the ERM proteins, ezrin and radixin may function as essential scaffold proteins responsible for the plasma membrane localization of PD-L1 in LS180 cells.

**Table 1 pharmaceuticals-14-00864-t001:** Primer sequences used for real-time quantitative reverse transcription-polymerase chain reaction (RT-qPCR) analysis of gene expression.

Gene	Primer Sequence (5′→3′)
h-*β-Actin* (forward)	TGGCACCCAGCACAATGAA
h-*β-Actin* (reverse)	CTAAGTCATAGTCCGCCTAGAAGCA
h-*PD-L1* (forward)	CAATGTGACCAGCACACTGAGAA
h-*PD-L1* (reverse)	GGCATAATAAGATGGCTCCCAGAA
h-*Ezrin* (forward)	ACCATGGATGCAGAGCTGGAG
h-*Ezrin* (reverse)	CATAGTGGAGGCCAAAGTACCACA
h-*Radixin* (forward)	GAATTTGCCATTCAGCCCAATA
h-*Radixin* (reverse)	GCCATGTAGAATAACCTTTGCTGTC
h-*Moesin* (forward)	CCGAATCCAAGCCGTGTGTA
h-*Moesin* (reverse)	GGCAAACTCCAGCTCTGCATC
h-*IFN-ɤ* (forward)	CTTTAAAGATGACCAGAGCATCCAA
h-*IFN-ɤ* (reverse)	GGCGACAGTTCAGCCATCAC
h-*TNF* (forward)	ACAACCCTCAGACGCCACAT
h-*TNF* (reverse)	GTGGAGCCGTGGGTCAGTAT
h-*IL-6* (forward)	AAGCCAGAGCTGTGCAGATGAGTA
h-*IL-6* (reverse)	TGTCCTGCAGCCACTGGTTC

## Data Availability

The datasets used and analyzed during this study are available from The Cancer Genome Atlas (https://www.cancer.gov/about-nci/organization/ccg/research/structural-genomics/tcga) and UALCAN (http://ualcan.path.uab.edu/analysis.html).

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
