# Peer review of "Role of Ezrin/Radixin/Moesin in the Surface Localization of Programmed Cell Death Ligand-1 in Human Colon Adenocarcinoma LS180 Cells"

_pharmaceuticals, 2021, doi:10.3390/ph14090864_

Round 1

Reviewer 1 Report

The content of the manuscript is timely. The manuscript is well written with detailed materials and methods, results, and a well-organized discussion section. I would give it an acceptance. 

What is the main question addressed by the research?

PD-L1 is an immune checkpoint protein that is highly expressed in cancer cells. A high expression of this protein is related to poor prognosis in colorectal cancer patients. Therefore the Kobori et al. tried to identify the key modulators for the cell membrane localization of PD-L1. They assumed that way, a novel therapeutic strategy for PD/PD-L1 blockade therapy could be developed.

Is it relevant and interesting?

The study is relevant since cancer immunotherapy is still not the greatest option for treating colorectal cancer patients. There is a dire need for immunotherapeutic strategies against this particular type of cancer.

How original is the topic?

To the best of my knowledge, the topic is novel and original.

What does it add to the subject area compared with other published material?

Although there have been studies to enhance the immunotherapeutic efficiency to target hard-to-target cancers, this work certainly adds up to the existing knowledge.

Is the paper well written?

The paper is very well written, and the text is clear and easy to read. I would still suggest for a minor grammar and english language check before publication.

Are the conclusions consistent with the evidence and arguments presented? Do they address the main question posed?

The conclusion is consistent with the evidence and arguments presented in the study. However, a little more detail on ‘how the study could impact the immunotherapeutic efficiency in Clinic’ would make the manuscript conclusive.

Author Response

Response to Reviewer 1 Comments

We would like to thank #Reviewer 1 for the greatest evaluation and valuable suggestions on our manuscript. We have carefully read all of your comments and suggestions and have made the corrections in the revised version of manuscript. Detailed responses to your comments are listed below, and we highlighted all changes with word track changes in the file labeled ‘Revised Manuscript with Track Changes’. We hope this revised manuscript would be satisfactory for publication in Pharmaceuticals.

Comment 1.  The conclusion is consistent with the evidence and arguments presented in the study. However, a little more detail on ‘how the study could impact the immunotherapeutic efficiency in Clinic’ would make the manuscript conclusive.

Reply Comments.

We would like to appreciate #Reviewer 1’s valuable suggestion. According to #Reviewer 1’s comment, we have added following sentence in the Conclusion section. If we did not understand the intent of your suggestion, please let us know.

Conclusion (Line 547 550)

This novel treatment strategy might give rise to possible adjuvant therapies for the current PD-1/PD-L1 blockade Abs, offering significant therapeutic benefits to CRC patients with intrinsic unresponsiveness and/or acquired resistance to the immune checkpoint therapies at present.

Reviewer 2 Report

Major

  1. Could authors discuss or explain more on why the moesin mRNA expression could not be detected in qPCR, while protein could be detected in Figure 1 d and 1 e?
  2. In Figure 4, the detected band of IP PD-L1 in the blots of Radixin and Ezrin are multiple bands which are not similar to the Input bands of Ezrin and Radixin. Could authors provide better blots on these 2 IP experiments?
  3. In Figure 6c, authors only did the flow cytometry work on Ezrin and Radixin siRNAs, how about Moesin siRNA?
  4. In the siRNA work of Moesin, authors used 2 nM instead of 5 nM. Could authors explain why this concentration was used instead of 5 nM? Moreover, in Figure 6d, if authors use 5nM Moesin siRNA instead of 2 nM, will the suppression of PD-L1 more insignificant as Radixin and Ezrin siRNAs?
  5. Authors study the correlation between PD-L1 and ezrin / radixin in human colon adenocarcinoma, how about moesin?
  6. Can authors discuss more about the above correlation in discussion part?
  7. As authors suggested that suppression of moesin at gene level might enhance production of certain cytokines which are key inducers of PD-L1 expression in tumor tissues, could authors also detect the cytokines expression levels too?
  8. May authors explain why 3 days of transfection for RNA work, while 4 days of transfection for protein and flow cytometry work?
  9. In method 4.6 protein isolation, could authors explain why 3 days of siRNA transfection was used instead of 4 days?
  10. In method 4.7 western blotting, Line 459, why HeLa cells were used which is totally not related to colon cancer?

Minor

  1. In figure 1e, could authors label which bands are LS180 cells and Caco-2 cells?
  2. In method 4.1 cell culture, could authors also add the type of antibiotics used?
  3. In method 4.4.1 Single immunofluorescence staining, Line 408, what does the meaning of “one drop”? Could author further clarify the volume used?
  4. In method 4.4.2, Line 426, could authors further clarify “one drop” definition? Moreover, what is the “ezrin, radixin, and moesin in blocking buffer”?

Author Response

Response to Reviewer 2 Comments

We would like to thank #Reviewer 2 for constructive opinion and valuable comments on our manuscript. We have carefully read all of your comments and suggestions and have incorporated them into the revised version of manuscript. Detailed responses to your comments are listed below, and we highlighted all changes with word track changes in the file labeled ‘Revised Manuscript with Track Changes’. We hope this revised manuscript would be satisfactory for publication in Pharmaceuticals.

Major Comments

Comment 1. Could authors discuss or explain more on why the moesin mRNA expression could not be detected in qPCR, while protein could be detected in Figure 1 d and 1 e?

Reply Comments.

As #Reviewer 2 pointed out, the moesin mRNA expression in Caco-2 cells could not detected in Figure 1d. On the other hands, in Figure 1e, the blotting bands in two lanes (duplicate) show each protein expression in LS180. Thus, is the moesin protein expression in Figure 1e represents only in LS180 cells. We are very sorry to have confused you. In order to avoid confusion for the readers, we incorporated the label of “LS180 cells” in Figure 1e.

Comment 2. In Figure 4, the detected band of IP PD-L1 in the blots of Radixin and Ezrin are multiple bands which are not similar to the Input bands of Ezrin and Radixin. Could authors provide better blots on these 2 IP experiments?

Reply Comments.

As #Reviewer 2 pointed out, we performed additional experiments to obtain better blots of Ezrin and Radixin co-immunoprecipitated with PD-L1 in Figure 4. To improve the quality of those blots as possible as we can, we replaced 5% non-fat dry milk with 5% bovine serum albumin in the blocking process and reduced the apply dose of total cell lysates (input) adjusted to protein concentrations at 0.20-0.25 µg/lane to optimize contrast of blots for Ezrin and Radixin co-immunoprecipitated with PD-L1. Based on the results, we revised blots of Ezrin and Radixin co-immunoprecipitated with PD-L1 in Figure 4.

Materials and Methods (Line 510 513)

The supernatant fractions and total cell lysates (input) were adjusted to protein concentrations ranging from 0.20 to 3.0 µg/lane, depending on the target protein, and separated via SDS-PAGE, which was followed by the electrotransfer onto a nitrocellulose membrane (Bio-Rad Laboratories).

Materials and Methods (Line 515 518)

The membrane was incubated in the blocking buffer containing 5% non-fat dry milk (FUJIFILM Wako Pure Chemical) for PD-L1, moesin, and β-actin or 5% BSA (FUJIFILM Wako Pure Chemical) for ezrin and radixin in PBS-T for 60 min at room temperature.

Comment 3. In Figure 6c, authors only did the flow cytometry work on Ezrin and Radixin siRNAs, how about Moesin siRNA?

Reply Comments.

As #Reviewer 2 pointed out, we incorporated the histogram for moesin siRNA into Figure 6c and revised the sentence in the corresponding Legend.

Legend for Figure 6 (Line 202 205)

(c) An overlay of the representative histograms for the mean fluorescence intensity of Alexa Fluor 488-labeled PD-L1 on the plasma membrane of LS180 cells treated with Lipofectamine (gray line), ezrin siRNA (pink line), radixin siRNA (green line), and moesin siRNA (orange line) as measured by flow cytometry.

Comment 4. In the siRNA work of Moesin, authors used 2 nM instead of 5 nM. Could authors explain why this concentration was used instead of 5 nM? Moreover, in Figure 6d, if authors use 5nM Moesin siRNA instead of 2 nM, will the suppression of PD-L1 more insignificant as Radixin and Ezrin siRNAs?

Reply Comments.

We thank for #Reviewer 2’s meaningful comments. The reason why we used moesin siRNA at the concentration of 2 nM instead of 5 nM is that treatment of LS180 cells with moesin siRNA at 5 nM significantly suppressed the cell viability in the preliminary experiments. In contrast, 5 nM of ezrin and radixin siRNA and 2 nM of moesin siRNA never influenced the cell viability of LS180 cells as shown in our previous manuscript below. In order to understand easier for the readers, we have incorporated the explanation how to decide the concentrations of each siRNA in this study and added the result of our preliminary experiment to check the influence of moesin siRNA concentrations at 2 and 5 nM on the cell viability of LS180 cells as Supplementary Figure 2.

Materials and Methods (Line 353 355)

The concentration of each siRNA and transfection reagent volume were determined to have a high knockdown activity and low cytotoxicity as shown in our recent publication [47] and Supplementary Figure 2.

Supplementary Figure 2 (Line 23 58)

Influence of Moesin siRNA Concentrations at 2 and 5 nM on the Cell Viability of LS180 Cells

Supplementary Figure 2. Influence of moesin siRNA concentrations at 2 and 5 nM on the cell viability of LS180 cells. Cells were treated with the transfection medium (Untreated), transfection reagent (Lipofectamine), nontargeting control (NC) siRNA, and specific siRNA for moesin and then incubated for 3 days. The concentrations of both siRNAs used were 2 and 5 nM. Cell viability was assessed with the PrestoBlue cell viability reagent. Staurosporine 1 μM was used as a positive control for inducing cell death. n = 3, ***p < 0.001 vs. Lipofectamine. All data were expressed as the mean ± SEM and analyzed by one-way ANOVA followed by Dunnett’s test.

Materials and Methods for Supplementary Figure 2

Cell Viability Assay

              LS180 cells were cultured until 70–80% confluent, and then were seeded at a density of 5.0 × 103 cells/well in 96-well cell culture plates (Thermo Fisher Scientific, Tokyo, Japan). The cultures were incubated overnight at 37°C in a humidified atmosphere with 5% CO2 to allow for attachment. Silencer Select small interfering RNA (siRNA) targeting human moesin or Silencer Select Negative Control siRNA (Thermo Fisher Scientific) were diluted with Opti-MEM (Thermo Fisher Scientific). The cells were then transfected with each siRNA (2 and 5 nM) using the Lipofectamine RNAiMAX Transfection Reagent (Thermo Fisher Scientific) at the volume of 0.2 μL/well. At the same time, cells were treated with staurosporine 1 μM (Merck, Darmstadt, Germany) as a positive control for inducing cell death. After treatment of cells with siRNAs or staurosporine, cells were continuously cultured for 3 days without exchanging medium. Thereafter, 10 µL/well of a commercial PrestoBlue Cell Viability Reagent (Thermo Fisher Scientific, Tokyo, Japan) was added directly to the wells containing 100 µL of the complete growth medium and the cells were incubated for 10 min at 37°C in a humidified atmosphere with 5% CO2 protected from direct light. Subsequently, fluorescence signals were detected at a wavelength of 560 nm (excitation) and 590 nm (emission) using a Synergy HTX Multi-Mode Microplate Reader (BioTek Instrument, Winooski, VT, USA). PrestoBlue is a new resazurin-based reagent to assess cell viability and cytotoxicity with higher sensitivity than 3-(4,5-dimethyl-2-thiazolyl)-2,5-diphenyl-2H-tetrazolium bromide (MTT), and comparable with that of Alamar Blue [1-4].

References for Supplementary Figure 2

  1. Xu, M.; McCanna, D. J.; Sivak, J. G., Use of the viability reagent PrestoBlue in comparison with alamarBlue and MTT to assess the viability of human corneal epithelial cells. J. Pharmacol. Toxicol. Methods 2015, 71, 1-7.
  2. Lall, N.; Henley-Smith, C. J.; De Canha, M. N.; Oosthuizen, C. B.; Berrington, D., Viability Reagent, PrestoBlue, in Comparison with Other Available Reagents, Utilized in Cytotoxicity and Antimicrobial Assays. Int. J. Microbiol. 2013, 2013, 420601.
  3. Stockert, J. C.; Horobin, R. W.; Colombo, L. L.; Blazquez-Castro, A., Tetrazolium salts and formazan products in Cell Biology: Viability assessment, fluorescence imaging, and labeling perspectives. Acta Histochem. 2018, 120, 159-167.
  4. Boncler, M.; Rozalski, M.; Krajewska, U.; Podsedek, A.; Watala, C., Comparison of PrestoBlue and MTT assays of cellular viability in the assessment of anti-proliferative effects of plant extracts on human endothelial cells. J. Pharmacol. Toxicol. Methods 2014, 69, 9-16.

Comment 5. Authors study the correlation between PD-L1 and ezrin / radixin in human colon adenocarcinoma, how about moesin?

Reply Comments.

As #Reviewer 2 pointed out, we incorporated the histogram for moesin siRNA into Figure 6c and revised the sentence in Results and Figure Legend.

Results (Line 215 218)

Gene correlation analysis of colon adenocarcinoma samples from TGCA detected a positive correlation between PD-L1 and ezrin (Pearson correlation coefficient: 0.36) or moesin (Pearson correlation coefficient: 0.42) (Figure 7a,b), but no correlation between PD-L1 and radixin.

Legend for Figure 7 (Line 220 222)

The correlation analysis of the gene expression of PD-L1 and (a) ezrin or (b) moesin was performed in clinical colon adenocarcinoma samples (n = 286) from The Cancer Genome Atlas (TGCA) database using the publicly available online tool UALCAN.

Comment 6. Can authors discuss more about the above correlation in discussion part?

Reply Comments.

As #Reviewer 2 pointed out, we incorporated the sentence into the Discussion section about the correlation analysis of the gene expression of PD-L1 and each ERM in the clinical tissue samples and also added more recent reports relevant to these results.

Discussion (Line 292 304)

Moreover, the gene correlation analysis revealed a positive correlation of PD-L1 with ezrin and moesin but not radixin in the clinical human colon adenocarcinoma samples from the TGCA database. Recent clinical retrospective study found that the positive PD-L1 expression rate in the CRC tissues was significantly higher in patients with lymph node metastasis than in those without lymph node metastasis, and also increased gradually as cancer stage became more advanced [32]. Furthermore, emerging evidence has demonstrated that during cancer development, the protein expressions of ezrin especially in the plasma membrane of cancer cells are elevated, leading to cancer progression, invasion, and metastasis [63-65]. These observations raise the possibility that differential expression of ERM proteins and PD-L1 may exists during cancer development. Collectively, the discrepancies between the basic and clinical studies may be, at least in part, due to the different ERM and PD-L1 expression profiles that depend on the cancer cell types and the clinical cancer stages.

References (Line 650 652; 726 - 732)

  1. Xu, J.; Zhao, W.; Liao, K.; Tu, L.; Jiang, X.; Dai, H.; Yu, Y.; Xiong, Q.; Xiong, Z., Clinical retrospective study on the expression of the PD-L1 molecule in sporadic colorectal cancer and its correlation with K-ras gene mutations in Chinese patients. Am. J. Transl. Res. 2021, 13, 6142-6155.
  2. Song, Y.; Ma, X.; Zhang, M.; Wang, M.; Wang, G.; Ye, Y.; Xia, W., Ezrin Mediates Invasion and Metastasis in Tumorigenesis: A Review. Front. Cell Dev. Biol. 2020, 8, 588801.
  3. Federici, C.; Brambilla, D.; Lozupone, F.; Matarrese, P.; de Milito, A.; Lugini, L.; Iessi, E.; Cecchetti, S.; Marino, M.; Perdicchio, M.; Logozzi, M.; Spada, M.; Malorni, W.; Fais, S., Pleiotropic function of ezrin in human metastatic melanomas. Int. J. Cancer 2009, 124, 2804-12.
  4. Brambilla, D.; Fais, S., The Janus-faced role of ezrin in "linking" cells to either normal or metastatic phenotype. Int. J. Cancer 2009, 125, 2239-45.

Comment 7. As authors suggested that suppression of moesin at gene level might enhance production of certain cytokines which are key inducers of PD-L1 expression in tumor tissues, could authors also detect the cytokines expression levels too?

Reply Comments.

As #Reviewer 2 pointed out, we have performed additional experiments to detect the alteration in the expression levels of three cytokines all of which are key inducer of PD-L1 expression in LS180 cells exposed to moesin siRNA. Based on the results of additional experiments, we revised the sentences in Discussion and Materials and Methods and incorporated the results into Supplementary Materials.

Discussion (Line 306 310)

The pro-inflammatory cytokines, such as interferon (IFN)-ɤ, tumor necrosis factor (TNF)-α, and interleukin (IL)-6, are well known to be key inducers for PD-L1 expression at the transcriptional level in tumor tissues [21]. In fact, moesin siRNA significantly increased the mRNA expression levels of TNF and IL-6 in LS180 cells, although those of IFN-ɤ were undetectable (Supplementary Figure 1).

Materials and Methods (Line 369 372)

The reactions were performed with the One Step TB Green Prime Script PLUS PT-PCR Kit (Takara Bio, Shiga, Japan) and the gene-specific primer for human PD-L1, ezrin, radixin, moesin, IFN-ɤ, TNF, IL-6, and β-actin (all purchased from Takara Bio) at a final concentration of 0.4 µM.

Supplementary Figure 1 (Line 13 22)

Gene Silencing of Moesin Upregulates the mRNA Expression Levels of Proinflammatory Cytokines in LS180 Cells

Supplementary Figure 1. Gene silencing of moesin upregulates the mRNA expression levels of proinflammatory cytokines in LS180 cells. Cells were incubated with the transfection medium (Untreated), transfection reagent (Lipofectamine), nontargeting control (NC) siRNA, and specific siRNAs for ezrin, radixin, or moesin and then cultured for 3 days. (a) Representative amplification curves of β-Actin (brown line), tumor necrosis factor (TNF) (red line), interleukin (IL)-6 (green line), and interferon (IFN)-ɤ (blue line; undetectable) mRNA expressions in the Untreated cells as determined by real-time quantitative reverse transcription-polymerase chain reaction. Gene expression levels of (b) TNF and (c) IL-6 mRNA normalized with β-actin in cells treated with each siRNA relative to that in cells treated with the transfection reagent alone. n = 3, ***p < 0.001, *p < 0.05 vs. Lipofectamine. All data were expressed as the mean ± SEM and analyzed by one-way ANOVA followed by Dunnett’s test.

Comment 8. May authors explain why 3 days of transfection for RNA work, while 4 days of transfection for protein and flow cytometry work?

Reply Comments.

We thank for #Reviewer 2’s meaningful comments. In terms of the siRNAs used in this study, treatment period for 3 days is sufficient to suppress the target mRNA expressions, while that for 4 days is necessary to effectively suppress the target protein expressions. Therefore, we adopted each treatment period in the siRNA experiments, and incorporated the explanation in the Materials and Methods.

Materials and Methods (Line 358 359)

Each treatment period adopted in this study was determined based on the manufacture’s protocol.

Comments 9. In method 4.6 protein isolation, could authors explain why 3 days of siRNA transfection was used instead of 4 days?

Reply Comments.

As Reviewer 2 pointed out, that was our listing mistake. Actually, 4 days is correct. Therefore, we have revised the Materials and Methods as follows. We are sorry for our mistake.

Materials and Methods (Line 461)

After treatment of cells with siRNAs for 4 days without exchanging medium,

Comments 10. In method 4.7 western blotting, Line 459, why HeLa cells were used which is totally not related to colon cancer?

Reply Comments.

As Reviewer 2 pointed out, that was our listing mistake. Actually, LS180 is correct. Therefore, we have revised the Materials and Methods as follows. We are sorry for our mistake too.

Materials and Methods (Line 469 470)

Briefly, to perform sodium dodecyl sulfate (SDS)-polyacrylamide gel electrophoresis (PAGE), total lysates of LS180 cells ------.

Minor Comments

Comment 1. In figure 1e, could authors label which bands are LS180 cells and Caco-2 cells?

Reply Comments.

As described in the Reply Comments to Major Comment 1, Figure 1e is the result of moesin protein expression only in LS180 cells. We are very sorry to have confused you. In order to avoid confusion for the readers, we incorporated the label of “LS180 cells” in Figure 1e.

Comment 2. In method 4.1 cell culture, could authors also add the type of antibiotics used?

Reply Comments.

We did not use any antibiotics for cell culture in this study according to the manufacture’s protocol provided by the European Collection of Authenticated Cell Cultures (ECACC).

Comment 3. In method 4.4.1 Single immunofluorescence staining, Line 408, what does the meaning of “one drop”? Could author further clarify the volume used?

Reply Comments.

As Reviewer 2 pointed out, “one drop” sounds a bit confusing to readers, therefore, we revised the sentence to clarify the volume of secondary antibodies used in Materials and Methods.

Materials and Methods (Line 414 419)

After washing thrice with PBS-T, the cells were incubated for 60 min at room temperature with an Alexa Fluor 488-conjugated goat anti-rabbit IgG (H+L) Ab (R37116; Thermo Fisher Scientific) or an Alexa Fluor 594-conjugated goat anti-rabbit IgG (H+L) Ab (R37117; Thermo Fisher Scientific) both prepared as dilutions by adding approximately 40 µL to 460 µL blocking buffer.

Comment 4. In method 4.4.2, Line 426, could authors further clarify “one drop” definition? Moreover, what is the “ezrin, radixin, and moesin in blocking buffer”?

Reply Comments.

As Reviewer 2 pointed out, “one drop” sounds a bit confusing to readers, therefore, we revised the sentence to clarify the volume of secondary antibodies used in Materials and Methods. In addition, the sentence “ezrin, radixin, and moesin in blocking buffer” #Reviewer 2 pointed out, was our listing mistake. Therefore, we have revised the Materials and Methods as follows. We are sorry for our mistake too.

Thank you again for giving us much detailed Review reports throughout our manuscript.

Materials and Methods (Line 434 437)

After washing thrice with PBS-T, the cells were incubated for 60 min at room temperature with an Alexa Fluor 594-conjugated goat anti-rabbit IgG (H+L) (R37117; Thermo Fisher Scientific) prepared as dilution by adding approximately 40 μL to 460 µL blocking buffer.

Round 2

Reviewer 2 Report

Authors have addressed most of the concerns from reviewer on revised manuscript. Here still has one more comment on the revised manuscript:

  1. Could authors show the cell viability test results of the concentrations of Ezrin and Radixin siRNAs to show why authors chose 5nM? For example, could authors perform the cell viability tests by using 2nM, 5nM and 10nM of both siRNAs?

Author Response

Response to Reviewer 2 Comments

We would like to thank #Reviewer 2 for constructive opinion on our manuscript. We have carefully read your comment and suggestion and have incorporated them into the revised version of manuscript. Detailed response to your comment is listed below, and we highlighted all changes with word track changes in the file labeled ‘Revised Manuscript with Track Changes’. We hope this revised manuscript would be satisfactory for publication in Pharmaceuticals.

Minor Comments

Could authors show the cell viability test results of the concentrations of Ezrin and Radixin siRNAs to show why authors chose 5nM? For example, could authors perform the cell viability tests by using 2nM, 5nM and 10nM of both siRNAs?

Reply Comments.

As Reviewer#2 pointed out, we incorporated the results of preliminary experiment to check the influence of Ezrin and Radixin siRNA at the concentrations of 2 and 5 nM on the viability of LS180 cells in Supplementary Figure 2. The reason why we have not used each siRNA at 10 nM is that each siRNA against Ezrin, Radixin, or Moesin at the concentration of 5 nM was sufficient to knockdown the expression levels of target gene as indicated in our recent publication (Kobori T. et al., PLoS One 2021, 16, e0250889).

Supplementary Figure 2 (Line 23 30)

Supplementary Figure 2. Changes in the cell viability of LS180 cells by treatment with ERM siRNAs at the concentrations of 2 and 5 nM. Cells were treated with the transfection medium (Untreated), transfection reagent (Lipofectamine), nontargeting control (NC) siRNA, and specific siRNA for ezrin, raidixn, or moesin and then incubated for 3 days. The concentrations of all siRNAs used were 2 and 5 nM. Cell viability was assessed with the PrestoBlue cell viability reagent. Staurosporine 1 μM was used as a positive control for inducing cell death. n = 3, ***p < 0.001 vs. Lipofectamine. All data were expressed as the mean ± SEM and analyzed by one-way ANOVA followed by Dunnett’s test.

Supplementary Figure 2 (Line 32 48)

Materials and Methods for Supplementary Figure 2

Cell Viability Assay

              LS180 cells were cultured until 70–80% confluent, and then were seeded at a density of 5.0 × 103 cells/well in 96-well cell culture plates (Thermo Fisher Scientific, Tokyo, Japan). The cultures were incubated overnight at 37°C in a humidified atmosphere with 5% CO2 to allow for attachment. Silencer Select small interfering RNA (siRNA) targeting human ezrin, radixin, or moesin, and Silencer Select Negative Control siRNA (Thermo Fisher Scientific) were diluted with Opti-MEM (Thermo Fisher Scientific). The cells were then transfected with each siRNA (2 and 5 nM) using the Lipofectamine RNAiMAX Transfection Reagent (Thermo Fisher Scientific) at the volume of 0.2 μL/well. At the same time, cells were treated with staurosporine 1 μM (Merck, Darmstadt, Germany) as a positive control for inducing cell death. After treatment of cells with siRNAs or staurosporine, cells were continuously cultured for 3 days without exchanging medium. Thereafter, 10 µL/well of a commercial PrestoBlue Cell Viability Reagent (Thermo Fisher Scientific, Tokyo, Japan) was added directly to the wells containing 100 µL of the complete growth medium and the cells were incubated for 10 min at 37°C in a humidified atmosphere with 5% CO2 protected from direct light. Subsequently, fluorescence signals were detected at a wavelength of 560 nm (excitation) and 590 nm (emission) using a Synergy HTX Multi-Mode Microplate Reader (BioTek Instrument, Winooski, VT, USA). PrestoBlue is a new resazurin-based reagent to assess cell viability and cytotoxicity with higher sensitivity than 3-(4,5-dimethyl-2-thiazolyl)-2,5-diphenyl-2H-tetrazolium bromide (MTT), and comparable with that of Alamar Blue [1-4].
